# Significance of Levocarnitine Treatment in Dialysis Patients

**DOI:** 10.3390/nu13041219

**Published:** 2021-04-07

**Authors:** Hiroyuki Takashima, Takashi Maruyama, Masanori Abe

**Affiliations:** Division of Nephrology, Hypertension and Endocrinology, Department of Internal Medicine, Nihon University School of Medicine, 30-1 Oyaguchi Kami-cho, Itabashi-ku, Tokyo 173-8610, Japan; takashima.hiroyuki@nihon-u.ac.jp (H.T.); maruyama.takashi@nihon-u.ac.jp (T.M.)

**Keywords:** carnitine, carnitine deficiency, end-stage kidney disease, peritoneal dialysis, hemodialysis

## Abstract

Carnitine is a naturally occurring amino acid derivative that is involved in the transport of long-chain fatty acids to the mitochondrial matrix. There, these substrates undergo β-oxidation, producing energy. The major sources of carnitine are dietary intake, although carnitine is also endogenously synthesized in the liver and kidney. However, in patients on dialysis, serum carnitine levels progressively fall due to restricted dietary intake and deprivation of endogenous synthesis in the kidney. Furthermore, serum-free carnitine is removed by hemodialysis treatment because the molecular weight of carnitine is small (161 Da) and its protein binding rates are very low. Therefore, the dialysis procedure is a major cause of carnitine deficiency in patients undergoing hemodialysis. This deficiency may contribute to several clinical disorders in such patients. Symptoms of dialysis-related carnitine deficiency include erythropoiesis-stimulating agent-resistant anemia, myopathy, muscle weakness, and intradialytic muscle cramps and hypotension. However, levocarnitine administration might replenish the free carnitine and help to increase carnitine levels in muscle. This article reviews the previous research into levocarnitine therapy in patients on maintenance dialysis for the treatment of renal anemia, cardiac dysfunction, dyslipidemia, and muscle and dialytic symptoms, and it examines the efficacy of the therapeutic approach and related issues.

## 1. Introduction

Carnitine, with a molecular weight of 161 Da, is a water-soluble quaternary amine. It is derived from lysine and methionine, which are two essential amino acids. Its primary role is in facilitating the transport of long-chain fatty acids to the mitochondrial matrix. These substrates are delivered for β-oxidation and the subsequent production of energy. Carnitine is primarily biosynthesized in the kidney and liver and is found in virtually all tissues but predominantly in cardiac and skeletal muscle.

Patients on hemodialysis often have carnitine deficiency [1]. Carnitine deficiency is associated with several clinical disorders, such as erythropoiesis-stimulating agent (ESA)-resistant anemia, muscle weakness, myopathy, and intradialytic muscle cramps and hypotension. Additional clinical disorders of carnitine deficiency include dyslipidemia, cardiac arrhythmia, cachexia, insulin resistance, and glucose intolerance [2,3,4]. The characteristic features of dialysis-associated carnitine deficiency are reduced levels of free carnitine and elevated levels of acylcarnitine. Free carnitine levels are mainly decreased by its removal during hemodialysis, whereas the accumulation of acylcarnitine and an aberrantly elevated plasma acylcarnitine to free carnitine ratio are due to deficient renal clearance and β-oxidation failure [1,2]. Accordingly, carnitine supplementation in dialysis patients with carnitine insufficiency may yield clinical benefits by ameliorating several of the above-mentioned conditions.

In this review, we describe the profile of carnitine metabolism and the effects of carnitine treatment on the metabolism and function of dialysis patients. We also assess the current findings related to the carnitine treatment of patients undergoing dialysis therapy, particularly its impact on cardiac function, ESA-resistant anemia, muscle symptoms, and malnutrition.

## 2. Carnitine Homeostasis

The main dietary sources of carnitine are meat products, with small amounts of carnitine found in vegetables [5,6]. About 100–400 mg per day of carnitine is provided from a normal diet. Dietary carnitine is absorbed from the intestine by both active and passive transport and meets 65–75% of daily needs. The remaining 25–35% is supplied by biosynthesis in the kidney and liver from methionine and lysine. Carnitine is found both intracellularly and extracellularly and in both non-esterified and esterified forms. The former is free carnitine, while the latter is acylcarnitines. Short-, medium-, and long-chain fatty acids are found in carnitine esters and are present in biological systems. The proportion of acylcarnitine varies widely according to physical activity, disease condition, and nutritional state. Under normal conditions in humans, acylcarnitine accounts for approximately 20% of total carnitine in serum, 10–15% of that in the liver and skeletal muscle, and 50–60% of that in urine [7,8,9].

Under physiological conditions, the total carnitine content in the body has been estimated to be 100 mmol. More than 90% of total body carnitine is found in skeletal muscle, with 2–3% in the liver and kidney. Thus, only 0.5–1% is present in the extracellular fluid [10]. The brain has a relatively low concentration of carnitine, despite being one of the few organs with endogenous biosynthesis capability. Carnitine cannot bind to protein and is mainly filtered at the glomeruli of the kidney. However, over 90% of filtered carnitine is reabsorbed by the proximal renal tubule in individuals with normal kidney function, and the serum excretory threshold level of free carnitine in the kidney appears to be 40 μmol/L, which is near the normal serum concentration of free carnitine [5,6]. Tubular reabsorption of free carnitine predominates. Therefore, the excretion of acylcarnitine by the kidney is 4- to 8-fold higher than that of free carnitine [5]. Plasma membrane transporters and carnitine-dependent enzymes are important for maintaining carnitine homeostasis. Together, free and acylcarnitine comprise the carnitine system.

The high-affinity Na^+^/carnitine cotransporter OCTN2 is the most physiologically associated plasma membrane transporter of carnitine [11]. OCTN2 is extensively found in numerous tissues, such as the heart, skeletal muscle, kidney, and placenta. OCTN2 is localized to the brush border of tubular epithelial cells in the kidney and is most active in the proximal tubules of the nephron, which is the site of approximately 65% of reabsorption and secretion [12]. The association of mutations in the OTCN2 gene with primary systemic carnitine deficiency indicates its importance [13].

Carnitine/acylcarnitine translocase (CACT) and carnitine acyltransferases are known as carnitine-dependent enzymes. CACT converts mitochondrial carnitine to cytoplasmic acylcarnitine and allows the flow of both carnitine and short-chain acyl-carnitines into and out of the mitochondria [14]. Carnitine acyltransferases exist in tissue-specific isoforms with distinct kinetic characteristics and with significant modulatory targets involved in fatty acid metabolism and coenzyme-A (CoA) release [15].

The proper function of OCTN2 and the various carnitine-dependent enzymes is needed to maintain the carnitine system. Carnitine has an important role in energy metabolism. It transports long-chain fatty acids across the inner mitochondrial membrane and modulates β-oxidation and the resulting adenosine triphosphate (ATP) production [16]. Furthermore, carnitine participates in intermediary metabolism by regulating the ratio of acyl-CoA/CoA in the cell. The main mechanisms underlying this function of carnitine are the production of short-chain acylcarnitines, which are catalyzed by carnitine acetyltransferase, and the conversion of carnitine to acylcarnitine, which is catalyzed by CACT [14,17]. Carnitine has a buffer action for accumulated acyl-CoA. The accumulation of acyl-CoA inhibits several enzymes, including acetyl CoA carboxylase, adenine nucleotide translocase, citrate synthetase, pyruvate dehydrogenase, and pyruvate carboxylase, and it induces mitochondrial dysfunction. Therefore, an accumulation of acyl groups within the mitochondria inhibits the activity of energy-producing enzymes. Acyl-CoA is restricted to the mitochondrial matrix and cannot pass the membrane. However, its acyl group is transferred from acyl-CoA to carnitine, and carnitine is metabolized into acylcarnitine. Acylcarnitine translocates from mitochondria to the extracellular fluid and is finally excreted via the urine. The detoxifying effects of carnitine are important for cell metabolism [18]. The fatty acid metabolism and functions of carnitine are shown in Figure 1.

Serum carnitine concentrations are 50–60 μmol/L, which is calculated as the sum of free carnitine and acylcarnitine. When serum-free carnitine drops below 20 μmol/L, the clinical symptoms of carnitine deficiency can develop. In patients with severe hereditary metabolic diseases, acylcarnitine is found in the serum and urine. In these patients, the endogenous carnitine pool falls into a deficit to manage the crucial acyl transfer, which increases the acyl/free carnitine ratio in serum. A ratio exceeding 0.4 has been used to indicate carnitine insufficiency in clinical practice [19]. Daily urinary total carnitine excretion typically consists of 50% acylcarnitine, resulting in a urinary acyl/free carnitine ratio of about 1.0 [20]. Carnitine homeostasis is shown in Figure 2.

## 3. Carnitine Deficiency in Patients Who Are Undergoing Dialysis Therapy

Carnitine homeostasis is profoundly perturbed in patients with end-stage kidney disease, particularly patients on dialysis. Dietary intake of carnitine is decreased due to falls in appetite, total energy levels, and protein intake. In addition, accumulating evidence has linked inflammation to malnutrition, and chronic inflammation might also interrupt carnitine transfer in the intestine [21]. Protein-energy wasting (PEW) and inflammation are the most pivotal risk factors for morbidity and mortality in patients on dialysis [22,23,24]. Carnitine biosynthesis can also fall in patients on dialysis due to reduced biosynthesis in the kidney and limited compensation by the liver [25]. Furthermore, the kidney disease may itself modulate OCTN2 activity on the renal tubule [26]. Filtered carnitine in the glomerulus cannot be reabsorbed in anuric patients undergoing hemodialysis. Therefore, chronic hemodialysis treatment reduces serum and tissue levels of carnitine and can promote acylcarnitine accumulation. As a result of the low molecular weight of carnitine and its high hydrophilicity and absence of protein binding, carnitine is significantly removed by the dialyzer [27,28].

According to Japanese guidelines [29], a free carnitine level < 20 μmol/L is defined as carnitine deficiency, a high risk of carnitine deficiency is defined as a level in the range of 20–36 μmol/L, and carnitine insufficiency is defined as a serum acyl/free carnitine ratio > 0.4. Consequently, serum carnitine levels are significantly lower in patients receiving hemodialysis than in healthy individuals at 22.0 ± 5.4 μmol/L and 43.3 ± 8.6 μmol/L, respectively [30]. Serum endogenous carnitine levels are significantly negatively correlated with dialysis therapy duration, with most of the reduction occurring within the first few months of hemodialysis initiation [30]. Long-term hemodialysis (i.e., longer than 1 year) is also linked to a marked 38% reduction in muscle carnitine pools compared with those before hemodialysis initiation [30]. Another investigation also reported that the total carnitine and acylcarnitine levels in muscle were significantly decreased in patients on dialysis [31]. We recently reported the prevalence of carnitine deficiency in 150 patients on hemodialysis [32]. Of these, serum free carnitine levels were below the normal range (36–74 μmol/L) in 90% of the patients, and 25.3% of the participants met the definition of carnitine deficiency (<20 μmol/L). Furthermore, 64.7% were diagnosed as having high risk of carnitine deficiency (acyl/free carnitine ratio > 0.4). In addition, just 13.3% of the participants (*n =* 20) had a normal ratio of ≤0.4 and 86.7% of the participants (*n =* 130) were diagnosed with carnitine insufficiency. A longer duration of dialysis was significantly associated with lower serum carnitine levels in multivariate analysis [32].

Acylcarnitine levels are significantly higher in patients on maintenance hemodialysis than in healthy individuals. Acylcarnitine levels are significantly elevated in patients who have been receiving hemodialysis for at least 12 months [4,28]. Indeed, acylcarnitine levels account for about 50% of the total serum carnitine stores in these patients compared with just 15% in healthy individuals [4,28]. Hemodialysis procedures decrease free, short-chain, medium-chain, and dicarboxylic acylcarnitines but do not affect long-chain acylcarnitines [33]. The dialytic removal of acylcarnitine during a single hemodialysis session is significantly associated with the carbon chain length of the acyl groups, with no major removal of the 18-carbon chain esters [34]. The removal rate of acylcarnitine clearly decreases as the carbon chain length increases because it increases their molecular weight and alters their lipophilicity. Furthermore, longer-chain acylcarnitines can bind to protein [35]. Therefore, the acyl/free carnitine ratio is positively correlated with the number of months on hemodialysis treatment [30,36]. Acylcarnitines are classified according to carbon chain length. Tandem mass spectrometry can determine the details of acylcarnitines, such as whether they are short-chain, middle-chain, and long-chain acylcarnitines. Tandem mass spectrometry has revealed that a lower ratio of acetylcarnitine (C2)/(palmitoylcarnitine + octadecenoylcarnitine [C16+C18:1]), which indicates the ratio of short-chain/long-chain acylcarnitines, in patients on hemodialysis is associated with all-cause mortality [37].

## 4. Removal of Carnitine by Dialysis Therapy

In 2018, 339,841 patients underwent maintenance dialysis in Japan. Of these, 37.0% were receiving hemodiafiltration. Approximately 71% of patients who were receiving hemodiafiltration were treated with online hemodiafiltration and the pre-dilution method [38,39]. Compared with conventional high-flux hemodialysis, hemodiafiltration is a more effective technique; it relies on high-flux membranes that can remove both small solutes, such as urea, and low-molecular weight proteins, such as β2-microglobulin [40,41]. Serum carnitine is removed by hemodialysis. Previous work determined the percent reduction in serum-free carnitine in patients on hemodialysis with or without diabetes and without levocarnitine treatment. The reductions in plasma free carnitine were −64.7% and −66.6% in patients with or without diabetes, respectively [33]. However, the hemodialysis procedure was not described in detail (i.e., blood and dialysate flow rates, treatment time, and Kt/V). We previously investigated the reduction rate of the serum carnitine level after single sessions of hemodialysis and hemodiafiltration [32]. Hemodialysis using high-flux dialyzers was conducted at blood and dialysate flow rates of 200–240 mL/min and 500 mL/min, respectively. Hemodiafiltration using high-flux hemodiafilters was performed at blood flow, replacement fluid, and dialysate flow rates of 200–300, 200–250, and 250–300 mL/min, respectively. Although no significant differences were evident in the patients’ baseline characteristics or in the pre-dialysis serum total, free, or acylcarnitine concentrations between the hemodialysis and hemodiafiltration groups, the Kt/V values were 1.28 ± 0.27 and 1.45 ± 0.31 in the hemodialysis and hemodiafiltration groups, respectively (*p* = 0.042). There was a significantly greater decrease in serum total, free, and acylcarnitine levels in the hemodiafiltration group. Reduction rates of serum free carnitine of 64% ± 4% and 75% ± 7% were obtained under hemodialysis and hemodiafiltration conditions, respectively (*p* < 0.0001). These findings indicate the greater clearance of small molecular weight solutes by hemodiafiltration.

Patients on peritoneal dialysis exhibit a decreased serum free carnitine level and increased acyl/free carnitine ratio compared with age- and sex-matched individuals with normal kidney function [42,43]. In patients on peritoneal dialysis, the mechanism of carnitine deficiency is considered to be decreased dietary intake of carnitine-containing food, decreased renal carnitine synthesis, and decreased renal excretion of acylcarnitine [27,30]. Another contributor might be the loss of free carnitine into the peritoneal dialysis fluid [44]. The prevalences of carnitine deficiency, high risk of carnitine deficiency, and carnitine insufficiency in peritoneal dialysis patients are comparable to those of age-, sex-, and dialysis vintage-matched hemodialysis patients [45]. Lower serum-free carnitine levels are associated with a longer duration of peritoneal dialysis and an older age.

## 5. Carnitine Supplementation in Dialysis Patients

The association between carnitine deficiency and a decreased serum-free carnitine level may result in various cellular metabolic disorders, such as reduced mitochondrial β-oxidation of fatty acids and consequent diminished energy production and storage of toxic acylcarnitines and suppression of carnitine-related enzymes involved in metabolism [46]. These carnitine-related metabolic aberrations may induce the above-mentioned clinical disorders frequently found in patients on dialysis, which include muscle weakness and cardiomyopathy, PEW, plasma lipid abnormalities, and ESA-resistant anemia, as well as hemodialysis-associated symptoms such as hypotension and muscle cramps [2,3,4].

Carnitine supplementation for the treatment of dialysis-related carnitine deficiency can be performed orally or intravenously. Multiple investigations have evaluated the benefits of carnitine supplementation in patients on dialysis. Intravenously administered levocarnitine has a bioavailability of 100%. When a dose of 1–2 g levocarnitine is intravenously administered to healthy individuals, the serum carnitine levels rapidly increase to 10 times that of the threshold for renal tubular reabsorption; 70–90% is consequently excreted in an unchanged form in the urine 12–24 h after administration. Therefore, a single dose of levocarnitine does not persist in the system for a sufficient length of time for any significant amount to equilibrate into the skeletal and cardiac muscle. However, for hemodialysis patients, an intravenous dose of levocarnitine remains in the blood for a long enough time for it to be taken up into the organs or tissue compartments, with up to about 90% of the administered levocarnitine possibly moved into tissues [4]. Chronic intravenous levocarnitine administration elevates muscle carnitine levels by between 60% and 200% [47,48,49,50].

In contrast, the bioavailability of oral levocarnitine administration is low, even in healthy individuals. Only 15% of a standard 2-g dose is absorbed into the blood in healthy individuals and just 5% of an oral 6-g dose [51,52]. The bioavailability of oral levocarnitine in patients on dialysis has not yet been evaluated. The metabolism of dietary carnitine and choline produces trimethylamine N-oxide (TMAO), which directly induces atherosclerosis in rodents [53,54]. Intestinal bacteria metabolize carnitine and choline to trimethylamine, which is absorbed in the intestine. Trimethylamine is itself oxidized by hepatic flavin monooxygenase to make TMAO [55]. Under normal conditions, TMAO is rapidly removed from the circulation, largely via excretion in the urine [56,57]. Accordingly, circulating TMAO levels appear to be associated with coronary artery disease and may also be associated with mortality in patients on long-term hemodialysis [58,59]. However, no study has shown whether oral levocarnitine treatment or TMAO levels would accelerate atherosclerosis in hemodialysis patients. Thus, additional work is required to evaluate the superiority, efficacy, and safety of intravenous levocarnitine administration compared with oral administration because there is no evidence of associations between the increased levels of TMAO and atherosclerosis progression in dialysis patients.

The National Kidney Foundation has stated that the detection and diagnosis of dialysis-related carnitine deficiency, as well as the decision to treat chronic dialysis patients with levocarnitine, should be determined by clinical symptoms and signs [60]. Furthermore, proof of decreased serum-free carnitine levels or an increased acyl/free carnitine ratio is dispensable for the clinical diagnosis of dialysis-related carnitine deficiency. Serum-free carnitine levels are helpful to rule out dialysis-related carnitine deficiency. However, low concentrations of serum-free carnitine cannot be used as a predictive factor of a clinical response to levocarnitine treatment.

In addition, the National Kidney Foundation has declared that the administration of levocarnitine to dialysis patients should be considered for the following four clinical conditions [60]: (1) patients with anemia who are unable to maintain optimal hemoglobin or hematocrit levels with the use of ESA, despite adequate iron status, and with no other identifiable cause of anemia or a hypo-response to ESA; (2) patients with intradialytic hypotension and no other possible causes with repeated symptomatic intradialytic hypotensive events requiring treatment; (3) patients with cardiomyopathy who have heart failure symptoms such as New York Heart Association class III–IV or symptomatic cardiomyopathy with documented impaired left ventricular ejection fraction (LVEF) and a poor response to standard medical therapy; and (4) selected patients who have symptoms that diminish their quality of life, including skeletal muscle weakness and malaise.

## 6. Anemia

In patients with end-stage kidney disease, anemia is induced by decreased production of erythropoietin by the kidney or fibrosis of the bone marrow. Renal anemia is commonly treated with ESA in patients with impaired kidney function. Although renal anemia strongly influences prognosis, higher-dose ESA may increase the risk of cardiovascular events in the dialysis population [61,62]. Moreover, the dosage of ESAs to maintain target hemoglobin levels varies widely among patients on dialysis [63]. A lower hematocrit level has been associated with shorter survival [64]. However, a lower hematocrit level was not a significant predictor of mortality in multivariate analysis adjusted for age, serum albumin, and the presence of diabetes. ESA resistance, which is characterized by inflammation and malnutrition, may be a significant novel predictor of mortality [64]. Patients with target hematocrit levels (i.e., 33–36%) receiving a higher ESA dose exhibit a rate of mortality double that of patients with hematocrit levels in the same range but receiving a low ESA dose. Therefore, the use of ESAs should be minimized and ESA resistance is recognized as an important marker for improving survival in the dialysis population.

Levocarnitine administration is suggested as a potential additional therapy to ESA in the management of renal anemia. The Centers for Medicare and Medicaid Services allow intravenous levocarnitine administration to patients on hemodialysis who have ESA-resistant anemia and decreased serum carnitine levels [64]. Although the most common cause of hyporesponsiveness to ESAs is iron deficiency, carnitine deficiency is proposed to be one of the causes of ESA-resistant anemia in Japanese Society for Dialysis Therapy guidelines [65]. Serum carnitine levels have been reported to be lower in patients with severe anemia needing high-dose ESA than in patients with mild-to-moderate anemia or no anemia [66]. In patients with a lower serum carnitine level and need for higher-dose ESA, erythrocyte membranes develop osmotic fragility. This shortens the survival time of erythrocytes and lowers hematocrit levels [67,68,69,70]. However, erythrocyte stability is reported to be improved by levocarnitine therapy, and this treatment would be associated with improved survival of erythrocytes through the following mechanism: levocarnitine regulates the erythrocyte membrane lipid complex, modifies the fatty acid metabolism, enhances the Na-K pump activity of erythrocytes, reduces membrane rigidity, and decreases erythrocyte calcium levels [69,71,72,73,74].

A systematic review and recent meta-analysis found that levocarnitine treatment ameliorates renal anemia and decreases ESA requirements in hemodialysis patients [75,76]. The efficacy of levocarnitine for treating renal anemia in patients on dialysis has been investigated by multiple studies. This work is summarized in Table 1 [43,69,70,71,77,78,79,80,81,82,83,84,85,86,87,88,89,90,91,92,93,94]. The aim of these studies was to maintain hematocrit or hemoglobin levels in carnitine and control groups by significantly decreasing the dosage of ESA in carnitine patients. ESA resistance can be determined by measuring the erythropoietin resistance index (ERI), which is calculated as the ESA dose divided by the hemoglobin level and body weight of each patient. This index is useful for assessing the response of the body to levocarnitine. Any decrease in ESA dosage or increase in the hemoglobin level during the observation period would decrease this index. The ERI was reduced by levocarnitine treatment in several studies, suggesting that levocarnitine improves erythropoietin efficiency versus control groups. However, the CARNIDIAL trial found no improvement in the ERI with levocarnitine administration in patients with a shorter duration of hemodialysis (<6 months) and no documented carnitine deficiency. In addition, levocarnitine treatment increased calcium and phosphate levels and was not associated with parathyroid hormone or fibroblast growth factor 23 [94,95].

Further studies should be conducted to determine whether levocarnitine treatment is effective in all dialysis patients with renal anemia and whether it improves long-term outcomes. Moreover, its dose–response profile in renal anemia has not yet been investigated.

## 7. Cardiac Function

Cardiovascular disease is a leading cause of mortality in dialysis patients [93]. Approximately 75% of end-stage kidney disease patients commencing hemodialysis treatment experience left ventricular dysfunction, represented by reduced LVEF, which is a significant risk factor for congestive heart failure [97]. Furthermore, intradialytic hypotension has been linked to mortality and is an independent predictor of mortality in this population [97,98,99].

The main energy source for cardiac myocytes is β-oxidation of fatty acids. Carnitine concentrations in myocytes are some of the highest of all cell types. Furthermore, the production of intracellular acylcarnitine and lactate is induced by myocardial ischemia. Thus, levocarnitine treatment might be useful for cardiac symptoms. Numerous investigations have reported the efficacy of levocarnitine treatment in terms of cardiac function; these are summarized in Table 2 [49,50,89,100,101,102,103,104,105,106,107,108].

The relationship between hypotensive episodes and levocarnitine treatment has also been investigated in dialysis patients. Patients who experience hypotension during hemodialysis treatment have lower serum carnitine levels than normotensive individuals [109]. Levocarnitine treatment significantly reduces intradialytic hypotension versus placebo [49,110]. Accordingly, intravenous levocarnitine supplementation is allowed for the management of dialysis-related hypotension in hemodialysis patients who have lower serum carnitine levels by the Centers for Medicare and Medicaid Services.

A strong correlation has been found between LVEF and serum carnitine levels in patients on dialysis. In addition, 3-month administration of levocarnitine improves LVEF, significantly so in patients with repeated hypotensive events [111]. It was suggested that patients experiencing symptomatic hypotension had a significantly lower LVEF and a higher mortality risk compared with asymptomatic patients [110]. Other studies have obtained similar results [89,103,112]. Mounting evidence favors a role for levocarnitine in the management of cardiac dysfunction. On the other hand, other studies have reported the ineffectiveness of levocarnitine treatment [50,104]; however, these findings must be interpreted with caution, because these studies included patients with normal LVEF. In our previous reports, atherosclerosis assessed by brachial-ankle pulse wave velocity and cardiac function assessed by LVEF and left ventricular mass index (LVMI) were improved by levocarnitine treatment in patients on hemodialysis [107,113]. Levocarnitine administration decreased N-terminal pro-brain natriuretic peptide (NT-proBNP) levels and ameliorated the ERI. Furthermore, the responders to levocarnitine treatment were patients with left ventricular hypertrophy, as defined by the LVMI on echocardiography. These results suggest that levocarnitine treatment might be effective for patients with a larger baseline LVMI [107]. Therefore, these results indicate that levocarnitine treatment is beneficial for patients with left ventricular hypertrophy, reduced LVEF, or dialysis-related hypotension.

Myocardial fatty acid metabolism, as assessed by 123-I–labeled β-methyl-p-iodophenyl-pentadecanoic acid (BMIPP), has been reported to be reduced in patients on long-term hemodialysis and recovered by levocarnitine therapy [102]. Tetradecyl glycidic acid (TDGA) impairs mitochondrial carnitine acyltransferase 1, and its administration induces left ventricular hypertrophy with enhanced lipid accumulation in the rat heart [111]. BMIPP washout from the myocardium is also decreased after TDGA administration [114]. Therefore, carnitine deficiency interrupts fatty acid metabolism in the myocardium and leads to myocardial lipid storage in patients on hemodialysis. A decreased free carnitine concentration results in disrupted fatty acid transfer into mitochondria; subsequently, the accumulation of acylcarnitine in the mitochondria disrupts carnitine-related enzymes involved in ATP production and transportation. Accordingly, levocarnitine treatment-induced amelioration of myocardial fatty acid metabolism and the acyl/free carnitine ratio might help to improve LVEF and decrease the LVMI.

Although levocarnitine treatment may be beneficial in improving LVEF, it is important to determine whether the treatment reduces cardiac events, hospitalizations, and mortality. To clarify the association between levocarnitine treatment and the hospitalization rate and number of hospital days, a large cohort study was conducted in patients on hemodialysis [115]. This study enrolled 2967 patients who were treated with levocarnitine for at least 3 months and had a 3-month or longer pre-levocarnitine period. The adjusted relative risk of hospitalization significantly decreased during the levocarnitine treatment compared with the rate before the initiation of levocarnitine treatment. Compared with the baseline hospitalization rate before levocarnitine treatment initiation, levocarnitine decreased the hospitalization rate by 34% and 58% at 6–9 months and 15–18 months, respectively. Furthermore, patients with cardiovascular disease, anemia, and hypoalbuminemia prior to levocarnitine treatment benefited most from levocarnitine treatment, in whom it was associated with fewer hospitalizations [115].

Uremia alters both carnitine and fatty acid metabolism. The combination of uremia-induced left ventricular hypertrophy and carnitine deficiency impairs myocardial metabolism and cardiac function. Levocarnitine treatment might partly improve the uremic hypertrophy, besides augmenting the metabolism. Additional large-scale clinical studies must be performed to clarify whether levocarnitine treatment ameliorates cardiovascular mortality in patients on dialysis.

## 8. Muscle Symptoms and Quality of Life

Sarcopenia and muscle weakness are frequent in patients with chronic kidney disease. Sarcopenia is caused by the aggravation of some physiological systems and is associated with aging. Decreased muscle strength and skeletal muscle mass are related to physical function [116,117]. In the general population, sarcopenia has been linked to adverse clinical outcomes, such as mortality, disability, hospitalization, falls, decreased quality of life, and need for long-term care [116,117]. Sarcopenia has also been associated with negative outcomes in patients with end-stage kidney disease or on dialysis [118,119,120,121]. Generally, physical activity falls with age in not only the general population, but also among patients with chronic kidney disease [122]. Patients on dialysis with decreased physical function have been found to have higher mortality than those with better physical function [123]. Although the clinical importance of sarcopenia is recognized, there are no clear intervention methods for the dialysis population. The pathophysiology of this syndrome is believed to be associated with amino acid deficiency, including that of carnitine.

In addition to sarcopenia, both inflammation and PEW are significant predictors of mortality in patients receiving dialysis therapy [22,23,24]. A recent meta-analysis reported a 28–50% prevalence of PEW or frailty in patients receiving dialysis [124]. Another report revealed that 30% of dialysis patients had mild or moderate malnutrition and that 6–8% of patients had severe malnutrition [125,126,127]. Although three pathophysiologies—sarcopenia, frailty, and PEW—are distinguished, they share some components that are associated with hospitalization and mortality. In particular, malnutrition and chronic inflammation complicated with sarcopenia are important predictors of clinical outcomes in patients on hemodialysis [128,129]. In addition, elevated proinflammatory cytokine levels stimulate protein catabolism through the ubiquitin–proteasome pathway, leading to muscle weakness or wasting [130]. The production of inflammatory cytokines, such as interleukin (IL)-1, IL-6, and tumor necrosis factor (TNF)-α, can be decreased by levocarnitine treatment [131,132,133].

Levocarnitine corrects insufficient energy supplies at the cellular level, alleviates long-chain fatty acid transport into mitochondria, and accelerates the removal of short- and medium-chain fatty acids stored during metabolism. Therefore, levocarnitine treatment may have beneficial effects on muscle wasting because fatty acid is the main source of energy in skeletal muscle [134]. Levocarnitine may increase the β-oxidation rate of fatty acids and maintain glycogen stores in skeletal muscle, thereby boosting ATP production [135]. Skeletal muscle function may be improved or maintained via levocarnitine-mediated augmentation of energy metabolism. Levocarnitine supplementation improves not only physical function but also mental and cognitive function in elderly individuals with normal kidney function [136,137]. Although levocarnitine supplementation fails to increase arm and leg muscle strength, it does increase the lean muscle mass of the arm and leg in elderly individuals with normal kidney function [138].

In Japan, patients receiving hemodialysis who had muscular symptoms such as cramps and asthenia have been found to have significantly lower endogenous serum carnitine levels compared with non-symptomatic patients [139]. Thirty patients on hemodialysis with muscular weakness, fatigue, or cramps were treated with levocarnitine for 12 weeks. Some muscle symptoms were improved in approximately 70% of the patients [139]. Fourteen patients on hemodialysis were treated with levocarnitine in a double-blind crossover manner to investigate carnitine levels in muscle and serum before and after 2 months of levocarnitine treatment. Although levocarnitine treatment ameliorated symptoms such as asthenia and cramps occurring during hemodialysis, these symptoms worsened during the washout period (i.e., after levocarnitine treatment was ceased) [140]. In addition, to evaluate the efficacy of levocarnitine for muscle function, a two-way parallel controlled trial was conducted for 6 months [141]. Muscle strength was significantly improved in four of the seven patients in the levocarnitine group at the study end, whereas none of the seven controls showed a significant improvement. Thereafter, all 14 patients were treated with levocarnitine for 10 months, with muscle strength increased in nine of the 14 patients. We previously conducted a randomized control trial of 91 hemodialysis patients who had lower serum carnitine levels [142]. The participants were randomly assigned to receive intravenous levocarnitine treatment (levocarnitine group) or no treatment (control group) for 12 months. Clinical dry weight, body mass index, and serum albumin levels fell significantly in the control group. However, there were no such results in the levocarnitine group. In addition, there were significant differences in the percent changes in arm muscle area, hand grip strength, and lean body mass after 12 months between the two groups [142]. Levocarnitine treatment was beneficial in patients on dialysis, particularly in elderly patients or those with diabetes, because it was able to maintain lean body mass and muscle function.

In addition to muscle and dialytic symptoms in patients on dialysis, a significant association has been reported between the acyl/free carnitine ratio and the physical component of the 36-Item Short Form Survey (SF-36) in men. Moreover, levocarnitine treatment improves SF-36 scores compared with baseline [68]. Furthermore, to evaluate health-related quality of life from the perspective of patients on dialysis, the SF-36 score was measured. Symptoms during hemodialysis were evaluated at each dialysis session using additional questionnaires. Six months of oral levocarnitine therapy boosted general health and physical function [143]. The efficacy of levocarnitine treatment for dialysis patients in terms of muscle symptoms, physical activities, and quality of life is summarized in Table 3 [49,50,83,84,87,100,139,140,141,142,143,144,145,146,147].

A meta-analysis failed to identify the clinical significance of levocarnitine treatment of intradialytic hypotension and muscle function [148]. However, some major limitations were noted, such as the small number of patients in many of the studies and a low associated statistical power. Furthermore, the definitions of dialysis-related hypotension and muscle cramps were not unified. To confirm the clinical efficacy of levocarnitine treatment of intradialytic hypotension and muscle cramps, additional adequately sized randomized clinical studies are required in this population.

## 9. Plasma Lipid Profiles and Inflammation-Related Parameters

Patients on dialysis exhibit a higher risk of atherosclerotic cardiovascular disease. Observational studies of dialysis patients have revealed a close relationship of dyslipidemia (e.g., elevated low-density lipoprotein (LDL) cholesterol, low high-density lipoprotein (HDL) cholesterol, elevated triglyceride, and/or elevated non-HDL cholesterol) with both atherosclerosis severity and risk of coronary artery disease [149,150]. Furthermore, dyslipidemia has a closer association with ischemic heart disease than with cerebrovascular disease. Several factors, including decreased activities of lipoprotein lipase and lecithin cholesterol acyltransferase (LCAT) and decreased hepatic lipase levels, promote dyslipidemia development in chronic kidney disease patients. The 2003 guidelines of the National Kidney Foundation’s Kidney Disease Outcomes Quality Initiative recommended a triglyceride level < 500 mg/dL in a fasting blood sample, an LDL cholesterol level < 100 mg/dL, and a non-HDL cholesterol level < 130 mg/dL [151].

Levocarnitine treatment may be beneficial for dyslipidemia in dialysis patients because carnitine increases the transport of free fatty acids into mitochondria and decreases the availability of free fatty acids for triglyceride synthesis. Decreased carnitine levels may be a possible contributing factor to hyperlipidemia in the dialysis population. In addition, carnitine treatment may improve dyslipidemia because carnitine stimulates the β-oxidation of long-chain fatty acids and decreases the ester bound to glycerol, even in dialysis patients [100].

Inflammation is highly prevalent in patients on hemodialysis, and elevated C-reactive protein is a predictor of all-cause and cardiovascular mortality in this population [152,153,154,155,156]. Inflammation can induce hepcidin overexpression and thus cause or aggravate absolute iron deficiency by inhibiting iron enteral absorption and functional iron deficiency through decreased release of stored iron from the liver and reticuloendothelial system [157]. The antioxidant and anti-inflammatory effects of levocarnitine have been described in vitro and in vivo [158,159]. Levocarnitine has also been shown to impact insulin sensitivity and protein catabolism; it has been proposed that increased levocarnitine is likely to improve nutritional status by reducing insulin resistance [160]. In one study, when patients were divided into two groups according to albumin level (<3.5 g/dL or ≥3.5 g/dL) before levocarnitine treatment, the higher albumin group displayed a significant increase in the prealbumin level and an improved malnutrition–inflammation score (MIS) [161]. Some clinical trials have indicated that levocarnitine supplementation can improve nutritional status in hemodialysis patients. It has been reported that oral levocarnitine supplementation tended to lower graft loss within 3 months after kidney transplantation, which might be related to the antioxidant effects of carnitine [162]. Several studies have examined the effects of levocarnitine treatment on plasma lipid levels and inflammation-related parameters in patients on maintenance dialysis. These studies are listed in Table 4 [48,80,85,86,88,90,92,131,132,133,161,163,164,165,166,167,168,169,170,171,172,173].

Multiple studies have shown that levocarnitine treatment has beneficial effects on dyslipidemia. Nonetheless, conflicting results were reported in some studies. A meta-analysis failed to identify beneficial effects of levocarnitine treatment on dyslipidemia in patients on dialysis [75,174]. However, another meta-analysis reported that levocarnitine administration decreased LDL-cholesterol levels in a subgroup of patients intravenously administered levocarnitine and with a longer interventional duration, whereas it was not associated with a reduction in total cholesterol and triglycerides levels or an increase in HDL-cholesterol levels [175]. Furthermore, meta-analyses demonstrated that levocarnitine administration decreased serum C-reactive protein levels in both statistically significant and clinically relevant manners [176] and that it increased total protein, albumin, transferrin, and prealbumin levels [177]. However, there were several limitations in previous studies, including differences among studies in plasma lipid levels and serum carnitine levels, levocarnitine dosage, administration methods, and study durations. Furthermore, research is required into specific dialysis populations with dyslipidemia, such as patients with low HDL cholesterol or high triglyceride levels.

## 10. Conclusions

The number of patients being treated with dialysis therapy is increasing worldwide. Patients with end-stage kidney disease who are receiving dialysis therapy frequently experience carnitine system dysfunction. Carnitine deficiency and uremic syndrome complicate the already complex pathophysiology of patients on dialysis. Furthermore, a dysfunctional fatty acid metabolism induces surplus production of free radicals and undesired apoptosis. Regarding carnitine deficiency, levocarnitine treatment positively affects pathologic processes in patients on dialysis. There are four principal indications for levocarnitine treatment in dialysis patients with carnitine deficiency according to the American National Kidney Foundation: (1) ESA-resistant anemia that has not responded to the standard ESA dosage; (2) recurrent symptomatic hypotension during hemodialysis; (3) symptomatic cardiomyopathy or confirmed cardiomyopathy with reduced LVEF; and (4) fatigability and muscle weakness that undermine quality of life. However, there were some limitations in the previous studies regarding levocarnitine treatment in the dialysis population, including sample size, adequacy of study design, and definition of target diseases. Furthermore, research has not been able to identify a dose–response relationship and the optimal administration route for levocarnitine treatment. Therefore, additional adequately sized clinical trials are required to determine whether levocarnitine treatment improves survival in patients on dialysis.

## Figures and Tables

**Figure 1 nutrients-13-01219-f001:**
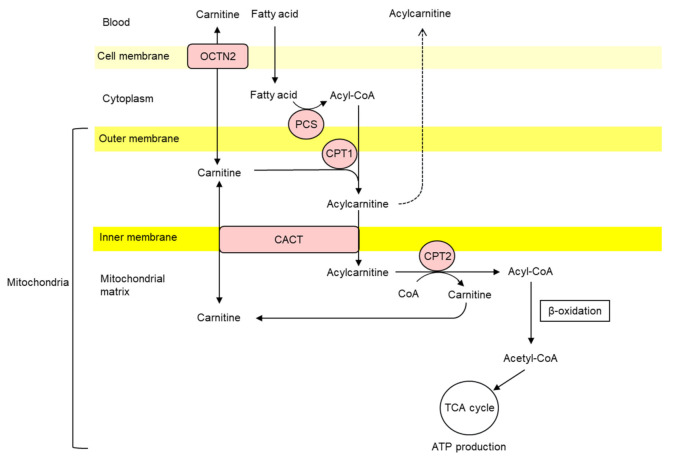
Fatty acid metabolism and metabolic functions of carnitine. CACT, carnitine acetyltransferase; CPT, carnitine palmitoyl transferase; OCTN2, organic cation/carnitine transporter 2, PCS, palmitoyl CoA synthetase.

**Figure 2 nutrients-13-01219-f002:**
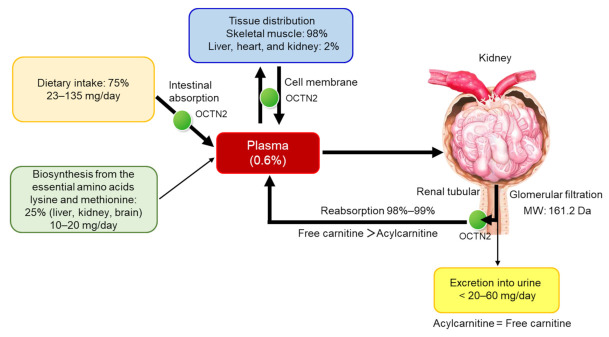
Carnitine homeostasis. MW, molecular weight; OCTN2, organic cation/carnitine transporter 2.

**Table 1 nutrients-13-01219-t001:** Studies of the effects of levocarnitine on renal anemia in dialysis patients.

Ref	Study Design	Subjects	Dose and Route	Treatment Duration	Findings ^a^
[77]	Two-way, parallel, double-blind	29 HD patients	20 mg/kg per Dx, IV	6 mo	↑ RBC survival T_0_: 39.1 days; T_6_: 42.7 days (*p* = 0.058)
		29 HD patients	Placebo, IV		→ RBC survival T_0_: 40.2 days; T_6_: 35.4 days (NS)
[78]	One-way, open-label	14 HD patients (ESA-resistant)	500 mg/day PO	3 mo	↑ Ht T_0_: 24.0% ± 2.0%; T_3_: 26.1% ± 2.0% (*p* = 0.003)
[71]	One-way, open-label	15 HD patients	30 mg/kg per Dx, IV	3 mo	↑ Ht T_0_: 30.8% ± 1.9%; T_3_: 34.2% ± 2.4% (*p* < 0.0001), ↓ Deformability of RBCs (*p* < 0.004)
[43]	One-way, open-label	12 PD patients	2 g/day PO	3 mo	↑ Ht T_0_: 35.4% ± 3.3%; T_3_: 38.1% ± 3.4% (*p* < 0.03), ↑ Hb T0: 11.0 ± 1.1 g/dL; T3: 11.9 ± 1.0 g/dL (*p* < 0.01)
[79]	Two-way, parallel, double-blind	28 HD patients	20 mg/kg per Dx, IV	6 mo	→ Ht T_0_: 34.1% ± 3.2%; T_6_: 32.8% ± 4.0% (NS)
		28 HD patients	Placebo		→ Ht T_0_: 32.9% ± 3.3%; T_6_: 33.9% ± 2.9% (NS)
	Four-way, parallel, double-blind	32 HD patients	10 mg/kg per Dx, IV	6 mo	→ Ht T_0_: 33.9% ± 3.2%; T_6_: 35.1% ± 4.2% (NS)
		30 HD patients	20 mg/kg per Dx, IV		→ Ht T_0_: 33.7% ± 3.5%; T_6_: 33.9% ± 3.4% (NS)
		32 HD patients	30 mg/kg per Dx, IV		→ Ht T_0_: 33.6% ± 3.3%; T_6_: 33.5% ± 2.7% (NS)
		33 HD patients	Placebo		→ Ht T_0_: 34.2% ± 3.2%; T_6_: 35.1% ± 4.2% (NS)
[80]	Two-way, parallel, double-blind	48 HD patients	20 mg/kg per Dx, IV	6 mo	↑ Hb T_0_: 9.7 ± 1.1 g/dL; T_6_: 10.8 ± 1.2 g/dL (*p* < 0.0001)
		65 HD patients	Placebo, IV		→ Hb T_0_: 9.8 ± 1.2 g/dL; T_6_: 9.9 ± 1.3 g/dL (NS)
[81]	Two-way, parallel, open label	78 HD patients	1 g/Dx, IV	7 mo	↑ Hb T_0_: 7.5 ± 1.5 g/dL; T_7_: 11.4 ± 1.2 g/dL (*p* < 0.05)
					↓ ERI T_0_: 183 ± 16 U/kg; T_7_: 142 ± 12 U/kg (*p* < 0.05)
		78 HD patients	No treatment		→ Hb T_0_: 7.5 ± 1.4 g/dL; T_7_: 9.2 ± 1.2 g/dL (NS)
					→ ERI T_0_: 185 ± 15 U/kg; T_7_: 160 ± 12 U/kg (NS)
[82]	Two-way, parallel, double-blind	18 HD patients	15 mg/kg per Dx, IV	6 mo	↑ Ht T_0_: 24.2% ± 2.2%; T_6_: 32.5% ± 3.7% (*p* = 0.001)
					↑ Hb T_0_: 7.9 ± 0.8 g/dL; T_6_: 10.3 ± 1.1 g/dL (*p* = 0.001)
		13 HD patients	Placebo, IV		→ Ht T_0_: 27.5% ± 4.5%; T_6_: 30.2% ± 4.0% (*p* = 0.1)
					→ Hb T_0_: 8.0 ± 0.4 g/dL; T_6_: 8.7 ± 2.5 g/dL (*p* = 0.4)
[83]	Two-way, parallel, single-blind	10 HD patients	20 mg/kg per Dx, IV	2 mo	↑ Hb +0.89 ± 0.56 g/dL vs. −0.47 ± 0.77 g/dL (*p* = 0.001)
		10 HD patients	Plaxevo, IV		
[84]	Double-blind, crossover, placebo-controlled	16 HD patients	20 mg/kg per Dx, IV	3 mo	→ ESA doses T_0_: 8562 ± 6762 U; T_3_: 8750 ± 7094 U (NS)
			Placebo, IV		→ Hb T_0_: 11.3 ± 1.9 g/dL T_3_: 11.5 ± 1.5 g/dL (NS)
[85]	Two-way, parallel, open-label	20 HD patients	1 g per Dx, twice a week, IV	6 mo	↑ Hb T_0_: 6.8 ± 1.0 g/dL; T_6_: 7.7 ± 1.1 g/dL (*p* < 0.001)
					↓ ERI values not reported (*p* < 0.001)
		20 HD patients	No treatment		→ Hb T_0_: 6.7 ± 1.0 g/dL; T_6_: 6.9 ± 1.0 g/dL (NS), → ERI (NS)
[86]	Two-way, parallel, open-label	20 HD patients	1 g/Dx, IV	3 mo	↑ Hb T_0_: 7.8 ± 1.3 g/dL; T_3_: 9.9 ± 1.9 g/dL (*p* < 0.05)
		20 HD patients	No treatment		→ Hb T_0_: 7.8 ± 1.1 g/dL; T_12_: 8.5 ± 1.2 g/dL (NS)
[87]	One-way, open-label	62 HD patients	600 mg/day, PO for 12 mo, then 1 g/Dx IV for 12 mo	24 mo	↑ Hb T_0_: 10.2 ± 1.2 g/dL; T_12_: 10.9 ± 0.9 g/dL
		18 PD patients	600 mg/day, PO	12 mo	→ Hb T_0_: 10.6 ± 1.1 g/dL; T_12_: 10.6 ± 1.3 g/dL
[88]	Two-way, parallel, double-blind	24 HD patients	1 g/day, PO	4 mo	→ Hb T_0_: 10.5 ± 2.5 g/dL; T_4_: 11.3 ± 2.1 g/dL (NS)
					↓ ESA doses T_0_: 7250 ± 5202 U/week; T_4_: 2500 ± 4180 U/week (*p* < 0.001)
		27 HD patients	Placebo, PO		→ Hb T_0_: 9.5 ± 2.2 g/dL; T_4_: 9.9 ± 2.5 g/dL (NS)
					↓ ESA doses T_0_: 8000 ± 3186 U/week; T_4_: 6000 ± 5083 U/week (*p* = 0.033)
[89]	Two-way, parallel, open-label	25 HD patients	1 g/Dx, IV and 1 g/non-Dx, PO	36 mo	↓ ESA doses T_0_: 5976 ± 1732 U/week; T_36_: 3391 ± 659 U/week (*p* < 0.001)
		35 HD patients	No treatment		→ ESA doses T_0_: 6100 ± 1587 U/week; T_36_: 5519 ± 1360 U/week (NS)
[90]	Two-way, parallel, double-blind	13 HD patients	20 mg/kg per Dx, IV	4 mo	→ ESA doses T_4_: −769 ± 1739 U/week (NS), → Hb T_4_: −0.08 ± 0.90 g/dL (NS)
		13 HD patients	Placebo, PIV		→ ESA doses T_4_: +153 ± 177 U/week (NS), → Hb T_4_: −0.26 ± 0.56 g/dL (NS)
[91]	Two-way, parallel, open-label	23 HD patients	15 mg/kg per Dx, IV	6 mo	→ ESA doses, → Ht (NS)
		22 HD patients	No treatment		
[92]	Two-way, parallel, double-blind	13 HD patients	1 g/Dx, IV	6 mo	↓ ERI T_0_: 102 ± 53 U/kg/week; T_6_: 63 ± 38 U/kg/week (*p* < 0.02)
		11 HD patients	Placebo, IV		→ ERI T_0_: 79 ± 32 U/kg/week; T_6_: 80 ± 47 U/kg/week (NS)
[70]	Two-way, parallel, double-blind	10 HD patients	1 g/Dx, IV	6 mo	↓ ERI T_0_: 135 ± 79; T_6_: 118 ± 108 U/kg per week per %Ht (*p* < 0.05)
		11 HD patients	Placebo, IV		↑ ERI T_0_: 136 ± 66; T_6_: 217 ± 204 U/kg per week per %Ht (*p* < 0.05)
[69]	Two-way, parallel, double-blind	20 HD patients	5 mg/kg or 25 mg/kg per Dx, IV	4 mo	↓ ERI T_0_: 16.0 ± 11.0; T_4_: 13.6 ± 10.5 U/kg per week per gHb (*p* < 0.02)
		20 HD patients	Placebo, IV		Values not reported
[96]	Two-way, parallel, double-blind	13 HD patients	20 mg/kg per Dx, IV	6 mo	↓ ERI -1.62 ± 0.91 vs. +1.33 ± 0.79 U/kg per gHb (*p* < 0.05)
		14 HD patients	Placebo, IV		
[93]	Two-way, parallel, open-label	30 HD patients	1 g/Dx, IV	12 mo	↓ ERI T_0_: 10.7 ± 7.3; T_12_: 6.4 ± 3.8 U/kg per gHb per week (*p* < 0.0001)
		30 HD patients	No treatment		→ ERI T_0_: 10.0 ± 7.9; T_12_: 9.6 ± 6.5 U/kg per gHb per week (NS)
[94]	Two-way, parallel, double-blind	46 HD patients	1 g/Dx, IV	12 mo	→ ERI T_0_: 20.6 ± 12.8; T_12_: 15.6 ± 15.9 IU/kg per gHb (*p* = 0.10)
		46 HD patients	Placebo, IV		→ ERI T_0_: 15.8 ± 11.3; T_12_: 9.5 ± 5.8 IU/kg per gHb (*p* = 0.10)

Dx, dialysis session; HD, hemodialysis; ERI, erythropoietin resistance index; ESA, erythropoiesis-stimulating agent; Hb, hemoglobin; Ht, hematocrit; IV, intravenous injection; mo, months; NS, not significant; PO, per oral; RBC, red blood cell; Ref, reference. ^a^ The findings show no difference (→), a decrease (↓), or an increase (↑).

**Table 2 nutrients-13-01219-t002:** Studies of the effect of levocarnitine on cardiac function and hypotension in dialysis patients.

Ref	Study Design	Population	Dose and Route	Treatment Duration	Findings ^a^
[101]	Two-way, crossover, double-blind	9 HD patients	990 mg/day PO then placebo for 2 mo each	2 mo	↓ Hypotension (*p* < 0.001)
		9 HD patients	Placebo then 990 mg/day PO for 2 mo each		→ Hypotension (NS)
[50]	Two-way, parallel, double-blind	14 HD patients	2 g/Dx, IV	6 weeks	No difference in cardiac function (NS)
		14 HD patients	Placebo		
[49]	Two-way, parallel, double-blind	38 HD patients	20 mg/kg per Dx, IV	6 mo	↓ Hypotension (*p* < 0.02)
		44 HD patients	Placebo		→ Hypotension (NS)
[102]	One-way, open-label	13 HD patients	1 g/Dx, IV	3 mo	↑ LVEF T_0_: 42.4 ± 19.4%; T_3_: 48.6 ± 17.6% (*p* < 0.05)
[89]	Two-way, parallel, open-label	25 HD patients	1 g/Dx, IV and 1 g/non-Dx PO	36 mo	↑ LVEF (*p* < 0.05)
		35 HD patients	No treatment		↓ LV end-diastolic volume (*p* < 0.05)
[103]	One-way, open-label	11 HD patients	1 g/day PO then 0.5 g/day PO for 1 mo each	2 mo	→ LVEDD, LVFS (NS)
					↑ Cardiac scintigraphy (*p* < 0.001)
[104]	One-way, open-label	9 HD patients (impaired LVEF)	500 mg/day, PO	6 mo	↑ LVEF T_0_: 44.9% ± 12.2%; T_6_: 53.8% ± 13.8% (*p* = 0.005)
					↓ CTR T_0_: 56.4 ± 5.4; T_6_: 53.8 ± 4.0 (*p* = 0.042)
[100]	One-way, open-label	11 HD patients (impaired LVEF)	1 g/Dx, IV	8 mo	↑ LVEF T_0_: 32.0% T_8_: 41.8% (*p* < 0.05)
[105]	Two-way, parallel, open-label	10 HD patients	10 mg/kg/day, PO	12 mo	↓ LVMI T_0_: 151.8 ± 21.2; T_12_: 134 ± 16 g/m^2^ (*p* < 0.01)
		10 HD patients	No treatment		→ LVMI T_0_: 153.3 ± 28.2; T_12_: 167.1 ± 43.1 g/m^2^ (NS)
[106]	Two-way, parallel, double-blind	20 HD patients	1500 mg/day, PO	6 mo	No difference in cardiac function (*p* = 0.67)
		35 HD patients	No treatment		Cardiac function was not investigated.
[107]	Two-way, parallel, double-blind	10 HD patients	900 mg/day, PO	3 mo	↑ LVEF T_0_: 61.8% ± 16.0% T_3_: 64.4% ± 13.8% (*p* < 0.05)
					↓ Hypotension T_0_: 4.0 ± 1.7; T_3_: 1.3 ± 0.9 times/mo (*p* < 0.05)
		8 HD patients	Placebo		→ LVEF (NS)
[108]	Two-way, parallel, open-label	75 HD patients	20 mg/kg/day, PO	12 mo	↑ LVEF T_0_: 53.1% ± 5.3% T_12_: 58.6% ± 5.5% (*p* < 0.001)
					↓ LVMI T_0_: 112 ± 26; T_12_: 107 ± 24 g/m^2^ (*p* < 0.001)
		73 HD patients	No treatment		→ LVEF, LVMI (NS)
[109]	Two-way, parallel, double-blind	18 HD patients	30 mg/kg/before Dx, IV	3 mo	↓ Hypotension 9.3% vs. 33.1% (*p* < 0.0001)
		15 HD patients	Placebo, IV		

CTR, cardiothoracic ratio; Dx, dialysis session; HD, hemodialysis; IV, intravenous injection; LVEDD, left ventricular end-diastolic dimension; LVFS, left ventricular fractional shortening; LVEF, left ventricular ejection fraction; LVMI, left ventricular mass index; mo, months; NS, not significant; PO, per oral; Ref, reference. ^a^ The findings show no difference (→), a decrease (↓), or an increase (↑).

**Table 3 nutrients-13-01219-t003:** Studies of the effect of levocarnitine on muscle symptoms and quality of life in dialysis patients.

Ref	Study Design	Subjects	Dose and Route	Treatment Duration	Findings ^a^
[101]	Double-blind, cross-over, placebo-controlled	18 HD patients	990 mg/day, PO Placebo, PO	2 mo	↓ Cramps (*p* < 0.001), ↓ Asthenia (*p* < 0.001), ↓ Dyspnea (*p* < 0.001)
[140]	Double-blind, cross-over, placebo-controlled	14 HD patients	2 g/day, PO Placebo, PO	2 mo	↑ Exercise time (*p* = 0.01), ↓ Asthenia (*p* = 0.01), ↓ Muscle cramps (*p* = 0.01)
[50]	Two-way, parallel, double-blindl	14 HD patients	2 g/Dx, IV	1.5 mo	No difference in muscular status (NS)
		14 HD patients	Placebo, IV		
[144]	One-way, open-label	6 HD patients	2 g/day, PO	1.5 mo	No difference in muscular function (NS)
[49]	Two-way, parallel, double-blind	38 HD patients	20 mg/kg per Dx, IV	6 mo	↓ Cramps (*p* = 0.02), ↓ Asthenia postdialysis (*p* = 0.04), ↑ O_2_ consumption (*p* = 0.03)
		44 HD patients	Placebo, IV		
[145]	One-way, open-label	26 HD patients	2 g/dialysate (*n =* 11), 2 g/day PO (*n =* 6), 2 g/Dx IV (*n =* 9)	6 mo	↓ Cramps (*p* = 0.04), ↓ Pain (*p* = 0.04), ↑ Isometric force (*p* = 0.001)
[146]	One-way, open-label	6 HD patients	2 g/day, PO	2 mo	↓ Cramps (*p* = 0.01), ↓ Weakness (*p* = 0.001), ↓ Fatigue (*p* = 0.05)
[139]	Two-way, parallel, open-label	30 HD patients	500 mg/day, PO	3 mo	↓ Weakness (*p* < 0.005), ↓ Fatigue (*p* < 0.005), ↓ Cramps/aches (*p* < 0.05)
		21 HD patients	No treatment		
[147]	Two-way, parallel, double-blind	9 HD patients	10 mg/kg per Dx, IV	4 mo	No difference in muscle cramps, uremic pruritus, physical strength, and general well-being
		8 HD patients	Placebo, IV		
[143]	Two-way, parallel, double-blind	101 HD patients	1 g/day, PO	6 mo	1.5 mo, ↑ QOL (*p* = 0.02); 3 mo, ↑ QOL (*p* = 0.015); >4.5 mo, ↓ QOL (*p* = 0.013)
			Placebo, PO		
[141]	Two-way, parallel, double-blind	7 HD patients	2 g/Dx, IV for 6 mo, then 1 g/Dx, IV for 10 mo	16 mo	→ Daily activity score T_0_: 3.5; T_6_: 2.0 (NS)
		7 HD patients	No treatment for 6 mo, then 1 g/Dx, IV for 10 mo		→ Daily activity score T_0_: 3.4; T_6_: 3.1 (NS)
[84]	Double-blind, cross-over, placebo-controlled	16 HD patients	20 mg/kg per Dx, IV	3 mo	No changes in muscle parameters and QOL scores
			Placebo, IV		
[96]	Two-way, parallel, double-blind	13 HD patients	20 mg/kg per Dx, IV	6 mo	↑ SF-36 scores T_0_: 33.9 ± 1.9; T_6_: 43.2 ± 3.0 (*p* < 0.05)
		14 HD patients	Placebo, IV		→ SF-36 scores T_0_: 40.6 ± 2.6; T_6_: 40.1 ± 3.0 (NS)
[83]	Two-way, parallel, single-blind	10 HD patients	20 mg/kg per Dx, IV	2 mo	↑ SF-36 scores T2: +18.3 ± 12.7 vs. −6.4 ± 16.4 (*p* = 0.001)
		10 HD patients	Placebo, IV		
[142]	Two-way, parallel, open-label	42 HD patients	1 g/Dx, IV	12 mo	↑ AMA: +2.11% vs. −4.11% (*p* < 0.01); ↑ LBM 0.70% vs. −2.22% (*p* < 0.001); ↑ HGS: +1.58% vs. −2.69% (*p* < 0.05)
		42 HD patients	No treatment		
[87]	One-way, open-label	62 HD patients	600 mg/day, PO for 12 mo, then 1 g/Dx IV for 12 mo	24 mo	↓ Muscle spasms in patients who had undergone HD for >4 years (*p*-value not reported)
		18 PD patients	600 mg/day, PO	12 mo	

AMA, arm muscle area; Dx, dialysis session; HD, hemodialysis; HGS, hand grip strength; LBM, lean body mass; IV, intravenous injection; mo, months; NS, not significant; PD, peritoneal dialysis; PO, per oral; QOL, quality of life; Ref, reference; SF-36, 36-Item Short Form Survey. ^a^ The findings showed no difference (→), or decrease (↓) or increase (↑).

**Table 4 nutrients-13-01219-t004:** Studies of the effect of levocarnitine on lipid profiles and inflammatory-related parameters in dialysis patients.

Ref	Study Design	Subjects	Dose and Route	Treatment Duration	Findings ^a^
[163]	Two-way, parallel, open-label	8 HD patients	0.5 g/Dx IV for 2 mo, then 1.0 g/Dx IV for 1.5 mo	3.5 mo	↓ TG T_0_: 336 ± 56 mg/dL; T_3.5:_ 244 ± 82 mg/dL (*p* < 0.05)
		8 HD patients	Placebo, IV	3.5 mo	→ TG T_0_: 329 ± 72 mg/dL; T_3.5_: 444 ± 82 mg/dL (NS)
[164]	Two-way, parallel, open-label	11 HD patients	1 g/Dx, IV for 1 mo then 2 g/Dx dialysate for 3 mo	4 mo	↓ TG, ↑ HDL (*p*-values not reported)
		11 HD patients	1 g/Dx, IV for 1 mo then 4 g/Dx dialysate for 3 mo		
[165]	Two-way, crossover, double-blind	9 HD patients	1 g t.i.d. PO then placebo for 5 wk each	5 wk	No difference in plasma lipid levels (NS)
		9 HD patients	Placebo then 1 g t.i.d. PO for 5 wk each	5 wk	
[48]	Two-way, parallel, double-blind	38 HD patients	20 mg/kg per Dx, IV	6 mo	No difference in plasma lipid levels (NS)
		44 HD patients	Placebo, IV	6 mo	
[166]	Two-way, parallel, double-blind	15 HD patients	1–1.5 g/Dx, IV	2 mo	No difference in plasma lipid levels (NS)
		15 HD patients	Placebo	2 mo	
[167]	Two-way, parallel, double-blind	11 HD patients	100 μmol/L dialysate	6 mo	No difference in plasma lipid levels (NS)
		10 HD patients	Placebo	6 mo	
[168]	Two-way, parallel, open-label	6 HD patients	900 mg t.i.d. PO	1 mo	↑ TG T_0_: 180 ± 66 mg%; T_1_: 219 ± 88 mg% (*p* < 0.05)
		4 HD patients	Placebo	1 mo	→ TG T_0_: 222 ± 35 mg%; T_1_: 222 ± 35 mg% (NS)
[131]	Two-way, parallel, open-label	21 HD patients	20 mg/kg per Dx, IV	6 mo	↓ TG T_0_: 1.6 ± 0.6; T_6_: 1.5 ±0.7 mmol/L (*p* = 0.001), ↑ TP T_0_: 6.4 ± 0.5; T_6_: 6.9 ± 0.5 g/dL (*p* < 0.001), ↑ Alb T_0_: 3.6 ± 0.3; T_6_: 4.1 ± 0.3 g/dL (*p* < 0.001), ↑ Tf T_0_: 1.2 ± 0.2; T_6_: 1.6 ± 0.4 g/L (*p* < 0.001), ↑ BMI T_0_: 23.4 ± 4.0; T_6_: 23.7 ± 4.0 (*p* < 0.001)
		21 HD patients	No treatment		→ TG, TP, Alb, Tf, BMI (NS)
[132]	Two-way, parallel, double-blind	20 HD patients	1 g/Dx, IV	6 mo	↓ CRP: T_0_: 2.1 ± 0.6 mg/dL; T_6_: 0.67 ± 0.1 mg/dL (*p* = 0.02), → TC, HDL, LDL, TG (NS)
		15 HD patients	No treatment		→ CRP, TC, HDL, LDL, TG (NS)
[88]	Two-way, parallel, double-blind	24 HD patients	1 g/day, PO	4 mo	↓ TG T_0_: 166 ± 71 mg/dL; T_4_: 138 ± 54 mg/dL (*p* = 0.001)
					↑ HDL T_0_: 30 ± 7 mg/dL; T_4_: 34 ± 7 mg/dL (*p* < 0.001)
		27 HD patients	Placebo, PO		↑ TG T_0_: 142 ± 58 mg/dL; T_4_: 151 ± 48 mg/dL (*p* = 0.029)
					→ HDL
[92]	Two-way, parallel, double-blind	13 HD patients	1 g/Dx, IV	6 mo	→ TC, HDL, TG (NS)
		11 HD patients	Placebo, IV		→ TC, HDL, TG (NS)
[90]	Two-way, parallel, double-blind	13 HD patients	20 mg/kg per Dx, IV	4 mo	→ TC, TG (NS)
		13 HD patients	Placebo, PIV		→ TC, TG (NS)
[169]	Two-way, parallel, double-blind	32 HD patients	600 mg/Dx, IV	12 mo	↓ MDA T_0_: 2.2 ± 0.7 μmol/mL; T_3_: 1.5 ± 0.7 μmol/mL (*p* < 0.001)
					↑ ABI T_0_: 0.71 ± 0.06; T_3_: 0.78± 0.08 (*p* < 0.001)
		32 HD patients	Placebo, IV		↑ MDA T_0_: 1.94 ± 0.5 μmol/mL; T_3_: 1.9 ± 0.7 μmol/mL (*p* < 0.01)
					↓ ABI T_0_: 0.75 ± 0.08; T_3_: 0.72 ± 0.01 (*p* < 0.001)
[85]	Two-way, parallel, open-label	20 HD patients	1 g/Dx, twice a week, IV	6 mo	↓ TC (*p* < 0.001),↑ HDL (*p* < 0.001), ↓ TG (*p* < 0.001)
		20 HD patients	No treatment		↑ TC (*p* < 0.001), ↓ HDL(*p* < 0.01), → TG (NS)
[86]	Two-way, parallel, open-label	20 HD patients	1 g/Dx, IV	3 mo	↓ TG T_0_: 190 ± 69 mg/dL; T_3:_ 179 ± 51 mg/dL (*p* < 0.05)
					↓ LDL 119± 21 mg/dL; T_3:_ 98 ± 19 mg/dL (*p* < 0.05)
					↓ CRP T_0_: 20.8 ± 1.7 μM; T_3_: 16.5± 1.3 μM (*p* < 0.05)
		20 HD patients	No treatment		→ TG, LDL, CRP (NS)
[161]	One-way, open-label	50 HD patients	1 g/Dx, IV	12 mo	↑ LDL (*p* = 0.005), ↓ HDL (*p* = 0.001), → TG (NS)
[133]	Two-way, parallel, open-label	18 HD patients	1 g/day, PO	3 mo	↓ CRP T_3:_ −1.6 ± 2.3 mg/L (*p* < 0.05), ↓ IL-6 T_3:_ −5.5 ± 3.6 ng/L (*p* < 0.001), ↓ IL-1β T_3_: −0.6 ± 0.6 ng/L (*p* < 0.001)
		18 HD patients	No treatment		→ CRP, IL-6, IL-1β (NS)
[170]	Two-way, parallel, double-blind	18 HD patients	1 g/day, PO	3 mo	↓ CRP T_0_: 7.5 ± 5.5 mg/L; T_3_: 4.4 ± 3.3 mg/L (*p* < 0.05)
		18 HD patients	Placebo, PO		→ CRP T_0_: 6.5 ± 5 mg/L; T_3_: 6.3 ± 3.1 mg/L (NS)
[171]	Two-way, parallel, double-blind	18 HD patients	1 g/day, PO	3 mo	↓ SAA T_3_: −32% (*p* < 0.001)
		18 HD patients	Placebo, PO		→ SAA (NS)
[172]	Two-way, parallel, open-label	17 HD patients	1 g/day, PO	3 mo	→ BMI, Leptin, Adiponectin (NS)
		25 HD patients	No treatment		→ BMI, Leptin, Adiponectin (NS)
[173]	Two-way, parallel, open-label	20 HD patients	1 g/day, PO	2 mo	→ Alb T_0_: 3.37 ± 0.40 g/dL; T_2_: 3.38 ± 0.43 g/dL (NS)
		20 HD patients	No treatment		→ Alb T_0_: 3.35 ± 0.34 g/dL; T_2_: 3.40 ± 0.38 g/dL (NS)
[80]	Two-way, parallel, double-blind	48 HD patients	20 mg/kg per Dx, IV	6 mo	↓ CRP T_0_: 1.8 ± 1.2 mg/dL; T_6_: 1.2 ± 0.2 (*p* < 0.002), ↑ Alb T_0_: 3.6 ± 0.3 g/dL; T_6_: 3.9 ± 0.4 g/dL (*p* < 0.0001), ↑ BMI T_0_: 20.5 ± 0.1; T_6_: 21.2 ± 0.5 (*p* < 0.0001)
		65 HD patients	Placebo, IV		→ CRP (NS), ↓ Alb (*p* < 0.0001), ↓ BMI (*p* < 0.05)

ABI, ankle brachial index; Alb, albumin; BMI, body mass index; CRP, C-reactive protein; Dx, dialysis session; HD, hemodialysis; HDL, high-density lipoprotein; IL, interleukin; LDL, low-density lipoprotein; MDA, malondialdehyde; IV, intravenous injection; mo, months; NS, not significant; PO, per oral; Ref, reference; SAA, serum amyloid A; TC, total cholesterol; Tf, transferrin; TG, triglyceride; TP, total protein. ^a^ The findings show no difference (→), a decrease (↓), or an increase (↑).

## Data Availability

Not applicable.

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
