# Peer review of "Significance of Levocarnitine Treatment in Dialysis Patients"

_nutrients, 2021, doi:10.3390/nu13041219_

Round 1

Reviewer 1 Report

In this paper, the authors provide an extensive review of carnitine metabolism in patients with chronic kindney disease, especially in those treated by dialysis.

After in depth review of carnitine physiology, the authors discuss the causes and consequences of carnitine insufficiency in CKD, and literature about substitutive therapy.

The paper is well written and well structured.

The tables, as presented in the manuscript, are not easy to read, especially the Findings columns. Studies could be separated by lines to help the reader to understand the findings. For most of these studies, statistical significance is not indicated, excepted for example , the placebo arm of study 79 in Table 1 (reported as NS).

As a general comment, the references cited by the authors are old and sometimes outdated. Among 153 references, only 14 are related to articles published in the last 5 years. Whereas it seems appropriated to cite seminal papers about physiology (cfr introduction), review of trials should be up to date and authors should prioritize large meta-analysis over old one-way open-label reports.

For exemple :

Table 1

Int Urol Nephrol . 2021 Mar 13. doi: 10.1007/s11255-021-02835-5. Effect of L-carnitine supplementation on renal anemia in patients on hemodialysis: a meta-analysis. Yan Zhu , Chao Xue , Jihong Ou , Zhijuan Xie  , Jin Deng PMID: 33713287

Table 4 (all references cited between 1980 and 1998 !)

Kidney Blood Press Res . 2013;38(1):31-41. Influence of L-carnitine supplementation on serum lipid profile in hemodialysis patients: a systematic review and meta-analysis. Haohai Huang

Other papers relating more recent meta-analysis to cite and discuss :

Am J Clin Nutr . 2014 Feb;99(2):408-22. doi: 10.3945/ajcn.113.062802. L-Carnitine supplementation for adults with end-stage kidney disease requiring maintenance hemodialysis: a systematic review and meta-analysis. Yizhi Chen  , Manuela Abbate, Li Tang, Guangyan Cai, Zhixiang Gong, Ribao Wei, Jianhui Zhou, Xiangmei Chen

J Nephrol . 2014 Jun;27(3):317-29. doi: 10.1007/s40620-013-0002-7. Effect of L-carnitine therapy on patients in maintenance hemodialysis: a systematic review and meta-analysis. Shi-Kun Yang

Biosci Rep . 2020 Jun 26;40(6):BSR20201639. doi: 10.1042/BSR20201639. The efficacy of L-carnitine in improving malnutrition in patients on maintenance hemodialysis: a meta-analysis. Jianwei Zhou 

Overall, carnitine supplementation in CKD on dialysis still  remains controversial. Nummerous studies have documented improvement of some parameters, but inconsistent results have also been published, especially in recent meta-analyses.

Although the authors recommend additional large-scale clinical studies at the end of each paragraph, studies publishing absence of positive clinical effect of carnitine supplementation are often omitted, or when mentioned, they are systematically associated with a comment inviting caution in the interpretation of the results (lines 342-344, 464-466, 494-498), giving the feeling of biased opinion in favor to carnitine supplementation.

Minor points :

Line 127 : unclear sentence : « Chronic hemodialysis treatment attempts to reduce serum and tissue carnitine levels » : attempts ? it is not a goal…

Line 290 : « with improved erythrocytes survival »

Reviewer 2 Report

The authors have presented a comprehensible review on the effects of carnitine deficiency in dialysis patients and highligthed the possibility of levocarnitine treatment to resolve this issue. The authors nicely and comprehensibly describe the studies that have been done so far and the limitations of the current work and conclude where more work is needed. With some minor editing and grammar corrections.

Reviewer 3 Report

In the present paper, Takashima et al review the subject of levocarnitine treatment in dialysis patients, providing a very thorough and up to date collection of the data available in the subject. In general, the review is very well written and clearly exposes the facts. I have just a few comments in order to try to improve the manuscript.

Major points

- The carnitine homeostasis section is very complete, but it will benefit from a figure that summarizes the main organs that participate in its metabolism and the pathways that follow. It might be even recommended that a second similar figure represents the impact of dialysis in those organs and pathways so anyone can have a clear picture with a glance at it.

- As glucose and lipid metabolism are strongly linked, there are data on the effect of carnitine on insulin-mediated pathways of glucose utilization and even protein metabolism that might be worth to be mentioned in a few lines. As diabetes is one of the main causes of ESRD a possible link could be built between them.

- Although very complete, some sections of the complications in dialysis in which this review is focused are missing some clinical papers on the subject that might be relevant to offer a more complete view.  For instance in the lipidic profile part, the authors should include recent papers from Katalinic et al (PMID: 30016788) and Ahmadi (27225722) and try to explain the results and the possible disagreements with the literature presented. There is also a metaanalysis by Chen et al (24368434) that should be included.

A paper showing the protective effect preventing intradialytic hypotension was also published in 2017 (28805348)

The same can be applied for the section on anemia. Please include the data on the CARDINAL trial (22935844). The same trial has published some information on other ESRD-associated pathologies like mineral metabolism (30408788), which could be also mentioned.

Finally a clinical trial on the effect of carnitine on allograft function in kidney transplant has been also carried out (28065453), a fact that should also be mentioned within the text.

Minor points

Please consider changing the following sentences

- Page 3 line 104: However its acyl group is transferred from acyl_CoA to carnitine, and carnitine is metabolized into acyl-carnitine.

- Page 3, line 127: Chronic hemodialysis treatment reduces serum and tissue carnitine levels

- Page 4 line 173: Compared with conventional high-flux hemodialysis, hemodiafiltration is a more effective technique

-Page 4 line 200: Another contributor might be the loss of free carnitine

Round 2

Reviewer 1 Report

The authors adequatly addressed most of concerns and the revised version of this manuscript is greatly improved and suitable for publication.